# Timing of antibiotic administration determines the spread of plasmid-encoded antibiotic resistance during microbial range expansion

Yinyin Ma [1,2] ✉, Josep Ramoneda[1,3] & David R. Johnson [1,4] ✉

Plasmids are the main vector by which antibiotic resistance is transferred between bacterial cells within surface-associated communities. In this study, we ask whether there is an optimal time to administer antibiotics to minimize plasmid spread in new bacterial genotypes during community expansion across surfaces. We address this question using consortia of *Pseudomonas stutzeri* strains, where one is an antibiotic resistance-encoding plasmid donor and the other a potential recipient. We allowed the strains to co-expand across a surface and administered antibiotics at different times. We find that plasmid transfer and transconjugant proliferation have unimodal relationships with the timing of antibiotic administration, where they reach maxima at intermediate times. These unimodal relationships result from the interplay between the probabilities of plasmid transfer and loss. Our study provides mechanistic insights into the transfer and proliferation of antibiotic resistance-encoding plasmids within microbial communities and identifies the timing of antibiotic administration as an important determinant.

The spread of antibiotic resistance (AR) is a global health problem whose causes and potential mitigation measures remain unclear[1,2]. The conjugation-mediated transfer of AR-encoding plasmids is a mechanism by which AR genes can spread between bacterial cells located within close spatial proximity to each other[3–6]. The frequencies of plasmid-free and plasmid-carrying cells within a microbial community will change over time depending on the probability of plasmid transfer from a plasmid donor to a potential recipient cell and the probability of plasmid loss upon cell division[7–13]. The frequencies will also depend on the relative fitness of plasmid-free and -carrying cells, where AR-encoding plasmids typically incur a fitness cost in the absence of antibiotic pressure[14–18]. The time during which the community is not exposed to antibiotic pressure is therefore expected to select against plasmid-carrying cells[16,19]. This leads to the expectation that a negative

relationship exists between the timing of antibiotic administration and the transfer and proliferation of AR-encoding plasmids in new genotypes, as longer times should result in smaller frequencies of plasmid-carrying cells due to out-competition by fitter plasmid-free cells.

In host-associated microbiomes, microbial communities often proliferate on surfaces (e.g. the gut lumen, skin, mucosae, etc.) where AR is typically conferred by conjugative plasmids[20,21]. AR in these systems can be maintained by plasmid transfer even in the absence of antibiotic pressure[22,23]. In patients receiving antibiotic treatment, these communities undergo frequent spatial reduction–expansion dynamics as a consequence of growth and death during which plasmid-free and plasmid-carrying individuals frequently (re)mix and expand together[24–26]. Work in the mouse gut has shown that the spread of AR-encoding plasmids is maximized in situations where pools of

[1]Department of Environmental Microbiology, Swiss Federal Institute of Aquatic Science and Technology (Eawag), 8600 Dübendorf, Switzerland. [2]Department of Environmental Systems Science, Swiss Federal Institute of Technology (ETH), 8092 Zürich, Switzerland. [3]Cooperative Institute for Research in Environmental Sciences (CIRES), University of Colorado, Boulder, CO 80309, USA. [4]Institute of Ecology and Evolution, University of Bern, 3012 Bern, Switzerland. ✉e-mail: yinyin.ma@eawag.ch; david.johnson@eawag.ch

persistent AR genotypes in the gut lumen mix with invading plasmid-free enteric pathogens[27,28]. It can be expected that the successional stage of these communities when antibiotics are applied can determine whether AR genotypes are likely to proliferate or not. The pervasiveness of mixed proliferation of plasmid-free and plasmid-carrying cells indicates that efforts to eradicate recalcitrant infections could benefit from a better temporal understanding of the spread of AR-encoding plasmids in relation to its main mechanisms of plasmid transfer and loss.

Surface-associated microbial communities, such as those associated with hosts, are considered hotspots for the conjugation-mediated transfer of AR-encoding plasmids[4,29,30], notably because surface association promotes the close physical cell–cell contacts that are required for the conjugation process[5,31]. A universal feature of surface-associated communities is that as cells within a community grow and divide, the community as a whole expands across space in a process referred to as range expansion[32–34]. During this process, growth is confined to only a thin layer of cells located at the expansion frontier where nutrients that diffuse from the periphery are readily available[35]. One consequence of this process is that different populations become increasingly spatially segregated over time[32,36–38]. This reduces the number of interspecific cell–cell contacts (e.g., between plasmid donors and potential recipients), thus also reducing the number of potential plasmid transfer events (Fig. 1a). Because spatial intermixing decays during range expansion and reduces the number of interspecific cell–cell contacts[32,36–39], this again leads to the expectation that a negative relationship exists between the time of antibiotic administration and the transfer and proliferation of AR-encoding plasmids in new genotypes.

In this study, we test the hypothesis that a negative relationship does indeed exist between the time of antibiotic administration and the transfer and proliferation of AR-encoding plasmids, where the negative relationship is driven by selection against plasmid-carrying cells in the absence of antibiotics and the decay in spatial intermixing during the range expansion process (Fig. 1a). Testing this hypothesis is especially paramount because, in clinical settings, infections generally need to be treated promptly, while our hypothesis would suggest that early treatment times might have negative consequences on the spread of AR-encoding plasmids in new genotypes (Fig. 1a). To test our hypothesis, we performed range expansion experiments with pairs of strains of the bacterium *Pseudomonas stutzeri*, where one strain carries the chloramphenicol resistance-encoding conjugative plasmid pAR145 (referred to as the plasmid donor strain) while the other is plasmid-free (referred to as the potential recipient strain) (Fig. 1b). After the initiation of range expansion, we applied chloramphenicol at different times and quantified the transfer and proliferation of pAR145. We then used an individual-based computational model to quantify how the probabilities of plasmid transfer and loss interact with each other to determine the spread of AR-encoding plasmids during range expansion. This enabled us to test the generality of our experimental results and establish a causal relationship between the timing of antibiotic administration and the spread of AR-encoding plasmids in new genotypes.

## Results

### pAR145 load declines in the absence of chloramphenicol

We first quantified the dynamics of pAR145 during range expansion in the absence of chloramphenicol, and thus in the absence of positive selection for pAR145. We defined the pAR145 load as the frequency of pAR145-carrying cells at the expansion frontier where cells are actively growing (approximately a radial ring with a width of 35 μm located at the expansion periphery[35]). We performed range

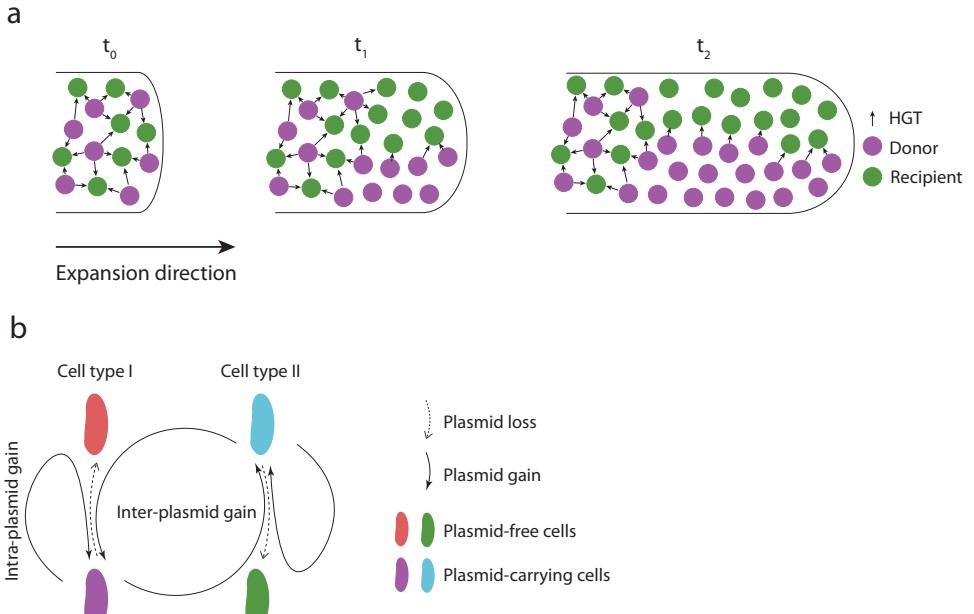

**Fig. 1 | Schematic of range expansion and experimental system used in this study. a** Different populations (in this case plasmid donors and potential recipients) become increasingly spatially segregated over time as a consequence of stochastic drift at the expansion frontier. This reduces the number of interspecific cell–cell contacts and the potential for plasmid transfer, as plasmid transfer can only occur along the interfaces of plasmid donors and potential recipients. **b** Our experimental system consists of pairs of strains of the bacterium *Pseudomonas stutzeri*. One strain is the plasmid donor that expresses red fluorescent protein from its chromosome and carries conjugative plasmid pAR145 that encodes for blue fluorescent protein and chloramphenicol resistance (Cell type 1; magenta cell). The other strain is the potential recipient that expresses green fluorescent protein from its chromosome and is plasmid-free (Cell type 2; green cell). If the potential recipient receives the plasmid, it will express both green and blue fluorescent proteins and appear in the composite color cyan. Plasmid carriers can also be cured of the plasmid during cell division and return to their plasmid-free states (magenta to red and cyan to green). Solid curved arrows indicate successful plasmid transfer while dashed curved arrows indicate plasmid loss. Inter-plasmid gain refers to plasmid transfer between different cell types, while intra-plasmid gain refers to plasmid transfer within the same cell type.

expansion experiments with consortia composed of two derivative strains of *P. stutzeri* A1601 (Fig. 1b). One expresses red fluorescent protein from its chromosome[40] and carries pAR145, which encodes for chloramphenicol resistance and blue fluorescent protein[41, 42] (referred to as the pAR145 donor). The other expresses green fluorescent protein from its chromosome[40] but does not carry pAR145 (referred to as the potential recipient). The pAR145 donor strain expresses both red and blue fluorescent proteins and appears as the composite color magenta, whereas the potential recipient only expresses green fluorescent protein and appears as green (Fig. 1b). If the potential recipient receives pAR145 (referred to as a transconjugant), then it will express both green and blue fluorescent proteins and appear as the composite color cyan (Fig. 1b). This system allows us to identify the spatial locations of pAR145 donors, potential recipients, and transconjugants, and to quantify the pAR145 load during range expansion.

We observed five important outcomes from the range expansion experiment. First, pAR145 donor and potential recipient cells rapidly segregated during range expansion to form a sectorized spatial pattern with reduced spatial intermixing (Fig. 2a, b), which is consistent with previous studies investigated pattern formation by other competing bacterial strains[32,37,38]. Second, abundant cyan sectors emerged during the early stages of sector formation (Fig. 2a), which

demonstrates extensive pAR145 transfer to potential recipient cells and the formation of transconjugants during the initial stages of range expansion when spatial intermixing was high. Third, the newly formed transconjugants were rapidly displaced by plasmid-free cells (green) thereafter (Fig. 2a, c). Fourth, extensive pAR145 loss occurred from pAR145 donor cells, which is evident by the rapid displacement of magenta sectors and the formation of red sectors (Fig. 2a and Supplementary Fig. 1a). Finally, carrying pAR145 incurs a fitness cost in the absence of chloramphenicol pressure (Supplementary Fig. 1b). This caused pAR145 donor cells to be gradually displaced by potential recipient cells (Fig. 2c), resulting in an increase in the ratio of green-to-red during range expansion (Fig. 2d). We also performed range expansion experiments with the pAR145 donor alone to verify that the decline in the pAR145 load was not dependent on the presence of the potential recipient (Supplementary Fig. 1a).

Overall, we observed a sharp decline in the pAR145 load during range expansion, where the decline began after ~24 h (corresponding to a radius of ~1750 μm) and the load approached zero after 48 h (corresponding to radii >2250 μm) (Fig. 2c). Our data indicate that the decline in the pAR145 load is caused by two processes. First, potential recipient cells that never received pAR145 (green cells) displaced pAR145-carrying cells, which is evident by the emergence and persistence of green sectors (Fig. 2a, d). Second, a subset of pAR145 donor

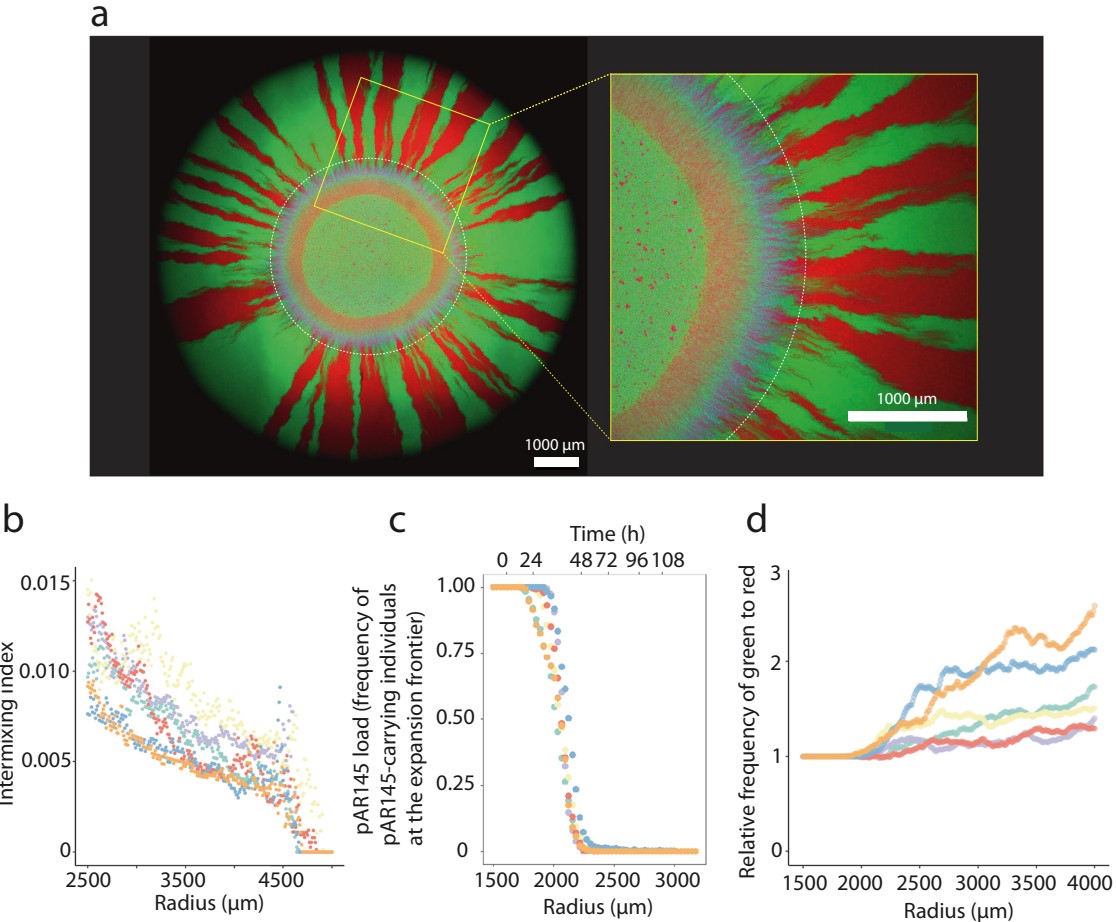

**Fig. 2 | pAR145 dynamics during range expansion in the absence of chloramphenicol pressure. a** Representative microscopy image of a one-week range expansion for a pair of *P. stutzeri* strains. One strain is the pAR145 donor that expresses red fluorescent protein from its chromosome and carries pAR145 encoding for blue fluorescent protein and chloramphenicol resistance (appears magenta). The other strain is the potential recipient that expresses green fluorescent protein from its chromosome. The white dashed ring indicates the boundary

between pAR145-carrying and largely pAR145-free regions. The yellow square frame is a magnified region. **b** Intermixing index, **c** pAR145 load, and **d** ratio of green (potential recipient) to red (donor cured of pAR145) pixels as a function of expansion radius (expansion time) beginning at the edge of the inoculation area (1500 μm) to the edge of the final expansion frontier (4000 μm) at radial increments of 10 μm. Different colored data points correspond to measurements for different independent biological replicates (*n* = 6).

cells lost pAR145 (red cells) and subsequently displaced pAR145-carrying cells, which is evident by the emergence and persistence of red sectors (Fig. 2a, c). Taken together, our data demonstrate that both the relative growth rates of pAR145-carrying and pAR145-free cells and the probability of pAR145 loss upon cell division are important for understanding and predicting pAR145 dynamics during range expansion.

### Administration time determines transconjugant proliferation

Because the pAR145 load rapidly declines during range expansion in the absence of chloramphenicol (Fig. 2c), we expected that the time at which chloramphenicol is administered after the onset of range expansion, and thus the time at which we apply positive selection for pAR145, determines the subsequent proliferation of transconjugant cells. More specifically, we hypothesized that the frequency of transconjugant cells at the expansion frontier would decline monotonically with time before applying chloramphenicol. To test this, we added chloramphenicol at 13 time points between 0 and 108 h after the onset of the range expansion experiment and allowed the consortia to expand thereafter for seven days. Thus, we fixed the chloramphenicol exposure time while varying the extent of range expansion prior to chloramphenicol administration. Note that the chloramphenicol concentration that we applied prevents further growth of plasmid-free cells. At the end of the experiment, we quantified the frequency of transconjugant cells at the expansion frontier.

Contrary to our expectation, we observed a unimodal relationship between the frequency of transconjugant cells at the expansion frontier seven days after chloramphenicol administration and the time at which we added chloramphenicol (two-sample two-sided Welch test; $P_1 = 6.3 \times 10^{-6}$, $P_2 = 8.8 \times 10^{-6}$, $n = 5$) (Fig. 3a, b). To test for a unimodal relationship, we computed $P_1$ and $P_2$ by comparing the maximum observed transconjugant frequency with the frequencies measured for chloramphenicol administration times at 0 h ($P_1$) and 108 h ($P_2$). Thus, we tested whether the maximum transconjugant frequency occurs at an intermediate chloramphenicol administration time. Furthermore, we found that the transconjugant frequency increased with the administration time up to the time at which we observed the maximum transconjugant frequency (Pearson correlation test; $r = 0.72$, $P = 8.8 \times 10^{-12}$, $n = 5$) and decreased thereafter (Pearson correlation test; $r = -0.81$, $P = 2.0 \times 10^{-10}$, $n = 5$) (Fig. 3b). At earlier chloramphenicol administration times, the spatial patterns that emerged after chloramphenicol administration consisted of contiguous discrete sectors of pAR145 donor and transconjugant cells (Fig. 3a). Thus, all cells that contributed to community expansion carried pAR145. At later chloramphenicol administration times (84–96 h), the sectorized patterns became discontiguous and were composed of spatially isolated bubble-like structures of pAR145 donor or transconjugant cells (Fig. 3a), presumably because only a few pAR145-carrying cells remained at the expansion frontier at the point when we administered chloramphenicol. We found that the level of spatial intermixing, which we reasoned is a determinant of pAR145 transfer, also has a unimodal relationship with the time of antibiotic administration (Fig. 3c).

We next quantified the extent of range expansion that occurred during the 7-day chloramphenicol treatment period (Fig. 3d). Because the frequency of transconjugant cells reached a maximum value when chloramphenicol was administered at intermediate times after the onset of range expansion (Fig. 3a, b), we also expected the ability of the consortia to expand (grow) during chloramphenicol treatment would also reach a maximum value at intermediate times. As expected, we observed a unimodal relationship between the extent of range expansion during chloramphenicol treatment and the time at which chloramphenicol was administered, with the maximum value occurring when chloramphenicol was administered 12 h after the onset of range expansion (two-sample two-sided Welch test; $P_1 = 0.011$, $P_2 = 1.5 \times 10^{-5}$, $n = 5$) (Fig. 3d). We also observed a significant positive

relationship between the frequency of transconjugant cells at the expansion frontier (data plotted in Fig. 3b) and the ability of the consortia to expand after the administration of chloramphenicol (data plotted in Fig. 3d) (Pearson correlation test; $r = 0.65$, $P = 1.5 \times 10^{-6}$, $n = 5$).

We finally quantified the extent to which pAR145 transferred into potential recipient cells by quantifying the ratio of transconjugant (cyan) to pAR145 donor (magenta) cells at the expansion frontier 7 days after chloramphenicol administration (Fig. 3e). We expected a positive relationship between the time of range expansion prior to chloramphenicol administration and the ratio of transconjugants-to-pAR145 donor cells. Briefly, short times before chloramphenicol administration should be insufficient to generate numerous transconjugant cells resulting in smaller ratios, while longer times should allow the generation of more transconjugant cells and higher accumulation of pAR145 donor cells that had lost pAR145 resulting in larger ratios. However, the ratio of transconjugant-to-pAR145 donor cells did not follow a monotonically increasing trend, but instead saturated at later chloramphenicol administration times (two-sample two-sided Welch test; $P = 0.17$, $n = 5$) (Fig. 3e). In this case, we computed $P$ by comparing the maximum observed ratio of transconjugant-to-pAR145 donor cells with the ratio measured at the longest chloramphenicol administration time (Fig. 3e).

### Local proliferation of individual transconjugant cells

We next sought to test whether the increased numbers of transconjugant cells at intermediate antibiotic administration times were due to more transconjugant cells being created (transfer events) or better proliferation of individual transconjugant cells. To test this, we performed individual-based computational simulations to gain insights using the CellModeller framework[43]. Briefly, we positioned plasmid donor (magenta) and potential recipient (green) cells at an ~1:1 ratio according to a checkerboard arrangement with a uniform distance between cells and random rotational orientation of cells along a two-dimensional plane. We assigned plasmid-free cells to have a 17% higher growth rate than plasmid-carrying cells in the absence of antibiotics, which is in accordance with our experimental data (Supplementary Table 1 and Supplementary Fig. 2). We applied a constant probability of plasmid transfer ($P_c = 0.002$) when a plasmid donor and a potential recipient cell come into physical contact with each other, whereupon successful plasmid transfer causes the recipient cell to become a transconjugant (cyan). We also applied a constant probability of plasmid loss upon cell division ($P_l = 0.005$) that can occur for any plasmid-carrying cell throughout the duration of range expansion. We administered antibiotics at various time steps (0, 100, 200, 400, 600, and 800) after initiating the simulations to mimic our experimental design, upon which only plasmid donor and transconjugant cells could continue growing. We used the same duration of "antibiotic exposure" for all simulations (1000 time steps) (Fig. 4a, b). Finally, we quantified the number of unique transconjugant lineages that derived from a single plasmid transfer event and persisted at the expansion frontier (Fig. 4c), the frequency of transconjugant cells at the expansion frontier (Fig. 4d), and the mean size of transconjugant lineages (Fig. 4e).

We observed a unimodal relationship between the number of transconjugant lineages and the time of antibiotic administration after the onset of range expansion (two-sample two-sided Welch test; $P_1 = 0.0012$, $P_2 = 0.00068$, $n = 5$) (Fig. 4c) and a unimodal relationship between the frequency of transconjugant cells at the expansion frontier and the antibiotic administration time (two-sample two-sided Welch test; $P_1 = 0.00058$, $P_2 = 0.00057$, $n = 5$) (Fig. 4d), which is consistent with our experimental observations (Fig. 3b). We also observed a unimodal relationship between the mean size of transconjugant lineages and the antibiotic administration time (two-sample two-sided Welch test; $P_1 = 5.6 \times 10^{-7}$, $P_2 = 0.00028$, $n = 5$) (Fig. 4e). Although the number of transconjugant lineages and the frequency of

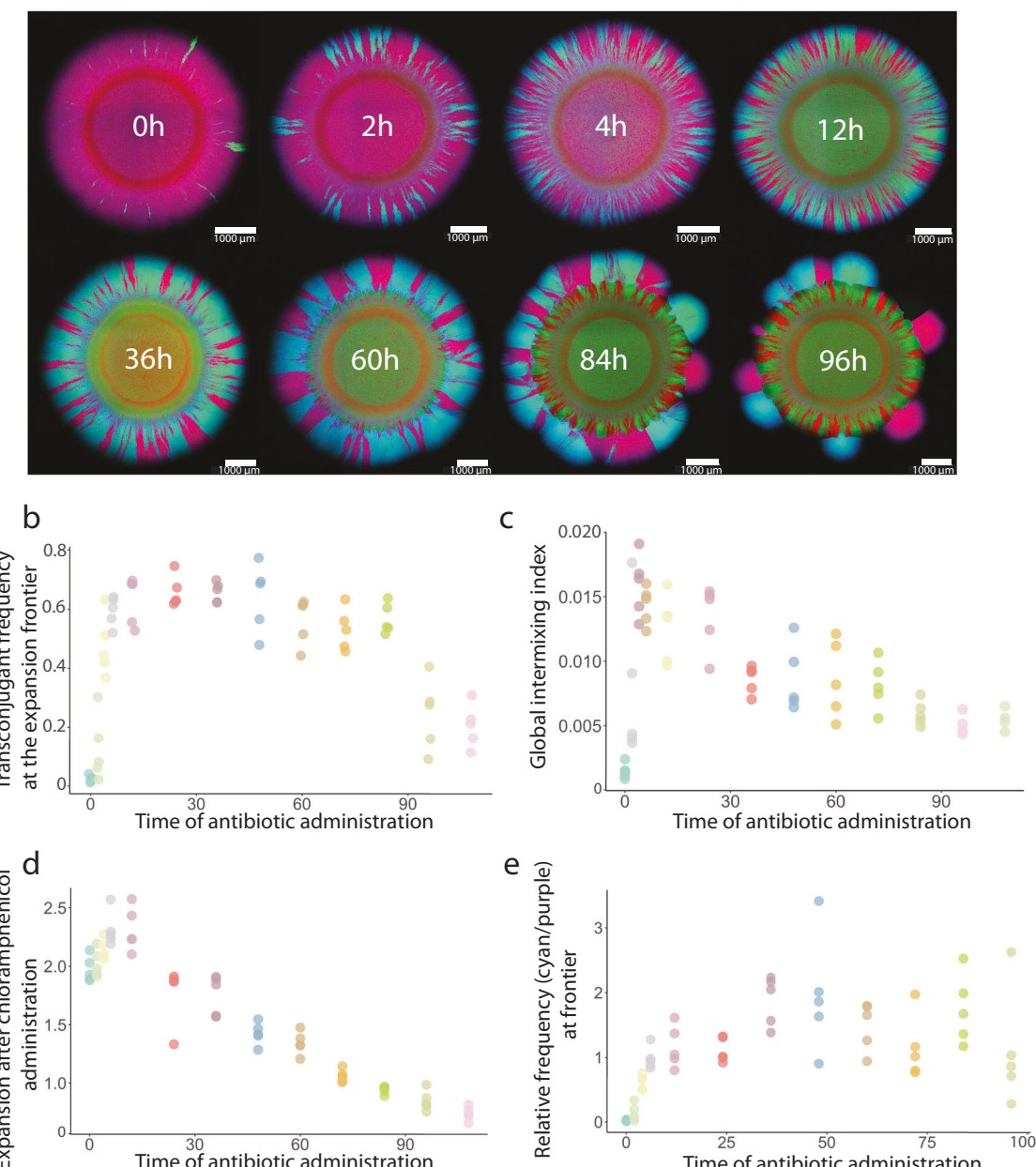

**Fig. 3 | Effect of the timing of chloramphenicol administration after the onset of range expansion on the spread of pAR145. a** Representative microscopy image (from $n = 5$) of spatial patterns formed after chloramphenicol was administered at different times after the onset of range expansion. The total time of chloramphenicol exposure was 7 days for all treatments. **b** Frequency of transconjugants (cyan) at the expansion frontier. **c** Global intermixing index measured as the sum of intermixing indices across the expansion area at radial increments of 10 μm. **d** Extent of range expansion after chloramphenicol administration (7 days). **e** Ratio of transconjugant (cyan) to pAR145 donor (magenta) cells at the expansion frontier. For **b**–**e**, each datapoint is a measurement for an independent biological replicate ($n = 5$).

transconjugant cells both show a unimodal relationship with the time of antibiotic administration, they do not correlate with each other (Spearman rank correlation test; rho = 0.32, $P = 0.085$). Instead, the mean size of transconjugant lineages is positively correlated with the frequency of transconjugant cells (Spearman rank correlation test; rho = 0.80, $P = 1.3 \times 10^{-7}$), indicating that the increased frequency of transconjugants is largely associated with local proliferation of individual transconjugants.

To assert that our results were independent of the time allowed for proliferation after antibiotic administration, we performed further simulations for an additional 1000 time steps after administering antibiotics and again quantified the number

of transconjugant cells at the expansion frontier (Supplementary Fig. 3a), the frequency of transconjugant cells at the expansion frontier (Supplementary Fig. 3b), and the mean size of transconjugant lineages (Supplementary Fig. 3c) for different antibiotic administration times. We found that the unimodal relationships remain valid after the additional time steps for all of these measurements. Moreover, we observed two additional outcomes. First, compared to the number of transconjugant lineages, the frequency of transconjugants at the expansion frontier increased with a longer duration of antibiotic exposure (two-sample two-sided Welch test; $P = 0.0044$ at time point 600) (Supplementary Fig. 3b). This is because as the antibiotic exposure duration

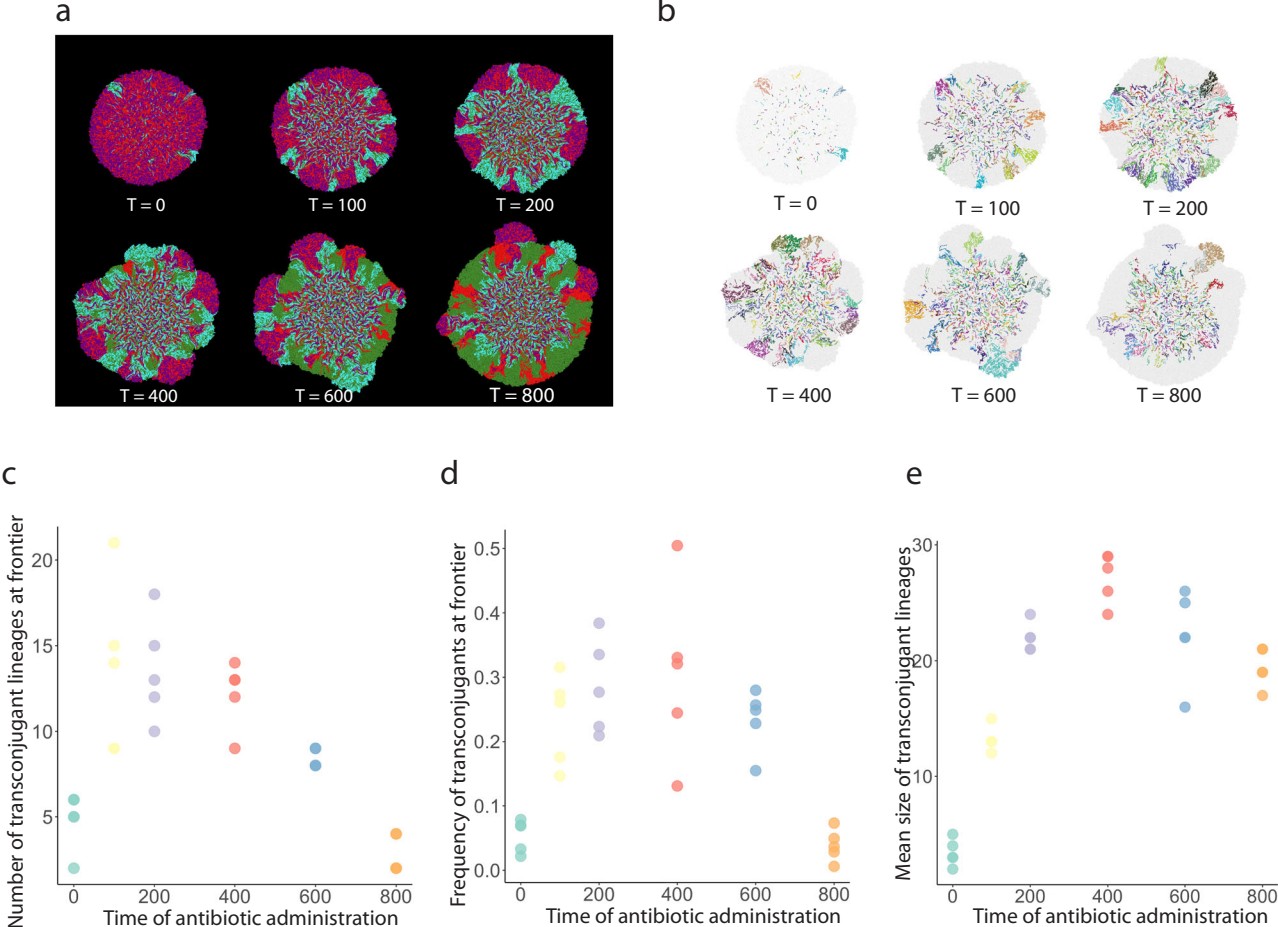

**Fig. 4 | Transconjugant lineage dynamics during range expansion.**
**a** Representative simulations of range expansions for different antibiotic administration times. Plasmid donor cells are magenta, transconjugant cells are cyan, potential recipient cells are green, and donor cells that lost the plasmid are red. T indicates the time step at which antibiotics were administered after the onset of range expansion. **b** Transconjugant lineages for the simulations presented in **a** where each color identifies a unique lineage. Effect of antibiotic administration time on **c** the number of transconjugant lineages at the expansion frontier, **d** the frequency of transconjugants at the expansion frontier, and **e** the mean size of transconjugant lineages. For **c**–**e**, each datapoint is a measurement for an independent simulation ($n = 5$).

increased, the "bubble-like" protrusions that we observed in both experiments and simulations gradually developed and merged together (Figs. 3a and 4a), therefore generating a higher frequency of transconjugants at the expansion frontier. However, because each protrusion originated from a single transconjugant lineage (Fig. 4b), the number of lineages remained constant. Second, the mean size of transconjugant lineages increased as the antibiotic exposure time prolonged (two-sample two-sided Welch test; $P = 0.0099$ at time point 600), which again verifies that the sizes of the newly formed transconjugant lineages can catch up provided there is a sufficiently long duration of antibiotic exposure.

**Interplay between plasmid transfer and loss probabilities**
We finally examined how the probabilities of plasmid transfer and loss, both of which can vary over orders of magnitude in nature[44–48], affect the relationship between the timing of antibiotic administration after the onset of range expansion and the frequency of transconjugant cells at the expansion frontier. More specifically, we tested under what conditions a unimodal relationship is likely to occur. To achieve this, we varied the plasmid transfer and loss probabilities in our individual-based computational model and quantified the effects. When we set the plasmid loss probability to a low value (0.001), we observed a monotonically increasing relationship where the frequency of

transconjugant cells at the expansion frontier increases with the plasmid transfer probability (Spearman rank correlation test; rho = 0.88, $P = 1.8 \times 10^{-10}$) (Fig. 5a, d), which is counter to our original expectation. This represents a scenario where the plasmid transfers at a faster rate than it is lost, thus ensuring its persistence in the system. In contrast, when we set the plasmid loss probability to a high value (0.015), we observed a monotonically decreasing relationship where the frequency of transconjugant cells at the expansion frontier decreases with the plasmid transfer probability (Spearman rank correlation test; rho = −0.63, $P = 0.00017$) (Fig. 5c, d), which is consistent with our original expectation. This represents a scenario where the plasmid is lost from the plasmid donor cells at a faster rate than the plasmid can transfer, eventually leading to it being purged from the system. Finally, when we set the plasmid loss probability to an intermediate value (0.005) (Fig. 5b), we observed a unimodal relationship where the frequency of transconjugant cells reaches a maximum at an intermediate antibiotic administration time (Fig. 5d), which is qualitatively consistent with our experimental observations (Fig. 3b). This represents a scenario where the plasmid transfer and loss processes are balanced and counteract each other.

We next expanded our modeling to investigate broader combinations of plasmid transfer and loss probabilities and quantified the resulting relationships between the time of antibiotic administration and the frequency of transconjugant cells at the expansion frontier.

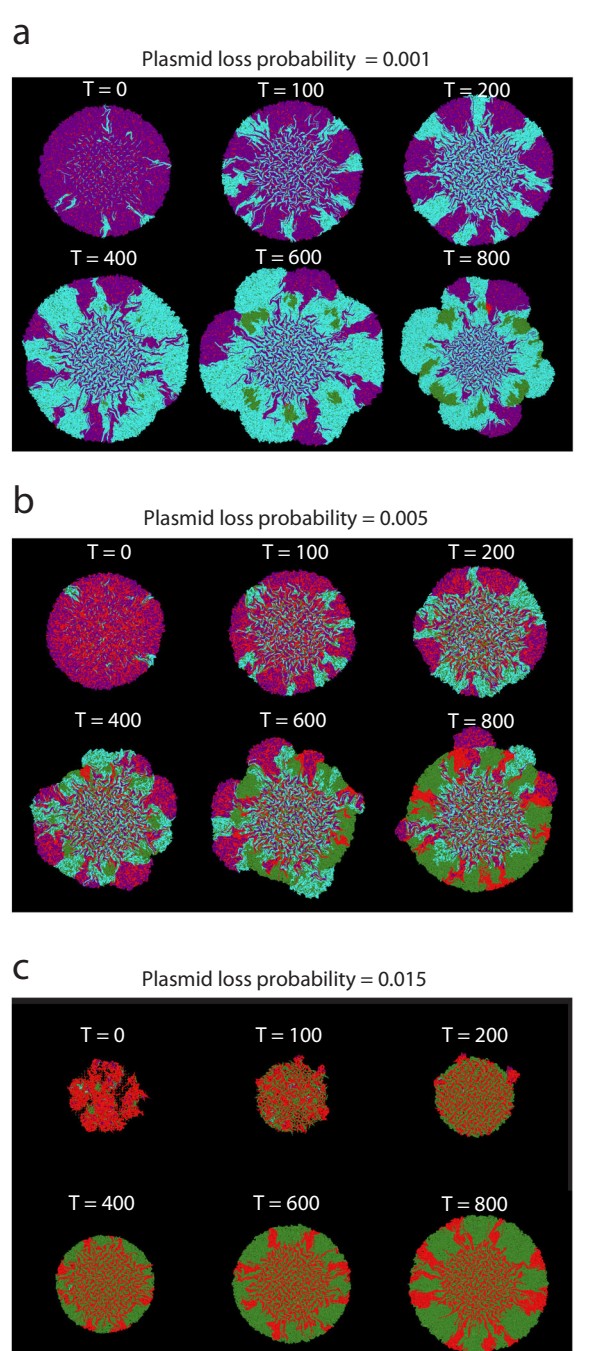

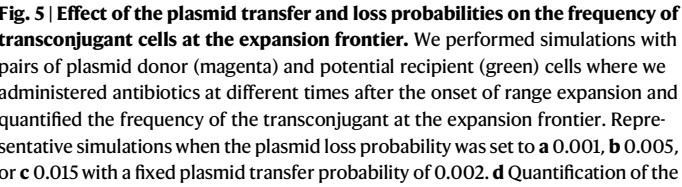

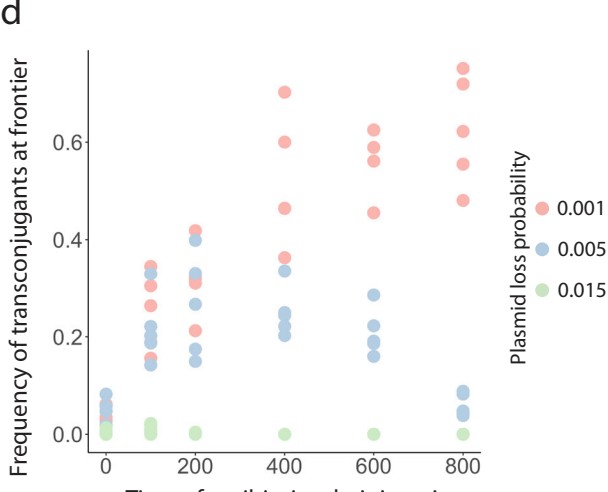

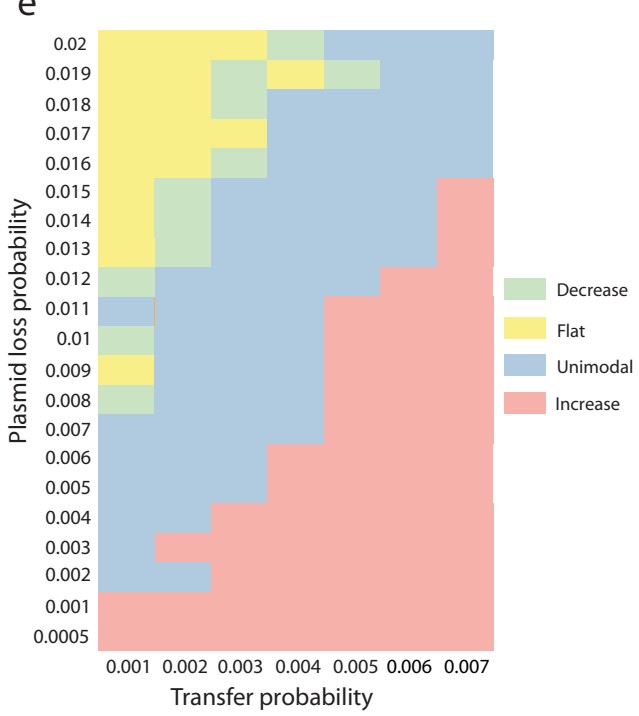

**Fig. 5 | Effect of the plasmid transfer and loss probabilities on the frequency of transconjugant cells at the expansion frontier.** We performed simulations with pairs of plasmid donor (magenta) and potential recipient (green) cells where we administered antibiotics at different times after the onset of range expansion and quantified the frequency of the transconjugant at the expansion frontier. Representative simulations when the plasmid loss probability was set to **a** 0.001, **b** 0.005, or **c** 0.015 with a fixed plasmid transfer probability of 0.002. **d** Quantification of the transconjugant frequencies at the expansion frontiers for (**a**–**c**). Datapoints are measurements for independent simulations (*n* = 5). **e** Quantification of the relationship between the time of antibiotic administration and the frequency of transconjugant cells at the expansion frontier for different combinations of plasmid transfer and loss probabilities. For each pair of plasmid transfer and loss probabilities and for each antibiotic administration time, we performed five simulations.

We varied the plasmid transfer probability from 0.001 to 0.007 and the plasmid loss probability from 0.0005 to 0.02 while fixing all other parameters. We then quantified the frequency of transconjugant cells at the expansion frontier. We found that combinations of plasmid transfer and loss probabilities can give rise to four distinct relationships (Fig. 5e). Monotonically increasing relationships (colored light red) occur for high plasmid transfer probabilities and

low plasmid loss probabilities, which is expected as such conditions enable prolonged plasmid persistence prior to antibiotic administration. Conversely, monotonically decreasing relationships (colored light green) occur for low plasmid transfer probabilities and high plasmid loss probabilities, which is again expected as such conditions reduce plasmid persistence prior to antibiotic administration. Flat relationships (colored yellow) can be considered as the extreme

case of monotonically decreasing relationships where the high plasmid loss probabilities purge plasmids before the earliest antibiotic administration time. Finally, unimodal relationships (colored light blue) lie at intermediate combinations of plasmid transfer and loss probabilities, which emphasizes that unimodal relationships emerge only when the plasmid transfer and loss processes are relatively balanced. Thus, predicting the spread of plasmid-encoded AR during range expansion requires knowledge of both plasmid transfer and loss probabilities.

## Discussion

Combining experiments with individual-based computational modeling, we demonstrated how the timing of antibiotic administration drives the spread of AR-encoding plasmids as surface-associated microbial communities expand across space. We showed that plasmid spread into AR-sensitive cells peaks at intermediate antibiotic administration times. These intermediate times are nested in a narrow window when the spatial intermixing of plasmid donors and potential recipients is maximal. The counterbalancing effects of plasmid transfer and loss predict the impact of the timing of antibiotic administration on the spread of AR.

In surface-associated microbial communities experiencing antibiotic pressure, the spread of plasmid-encoded AR is maximized for patterns of spatial organization displaying large numbers of contacts between plasmid donors and recipients (Fig. 3). The emergence of such patterns is dependent on the successional stage of the community; at early stages of community development the expansion frontier becomes rapidly dominated by AR types due to a fitness advantage over sensitive individuals (Fig. 3a). At late stages of community development, the long period of antibiotic-free conditions allows sensitive cells to dominate the expansion frontier due to the fitness cost derived from plasmid maintenance in their AR counterparts[49] (Fig. 3a). Antibiotic administration at a late stage of community development, therefore, occurs after the purging of plasmid-encoded AR and is expected to be the point when the community is most vulnerable to antibiotic stress. This vulnerability does not imply that late antibiotic administration times can completely eradicate the microbial community of interest[50,51], but it marks a point when the active AR fraction is at a minimum. Intermediate stages of community succession have allowed the preferential proliferation of plasmid-free individuals without completely outcompeting the AR fraction. This maximizes the contacts between plasmid-carrying and plasmid-free cells that promote plasmid transfer and results in the maximal spread of plasmid-encoded AR (Fig. 3a). The frequent mixing and proliferation of plasmid-carrying and plasmid-free populations of enterobacterial pathogens is considered an important factor of AR persistence and spread in the gut lumen[25]. A temporal perspective on plasmid-encoded AR spread in the gut could thus improve the understanding of the processes leading to recalcitrant AR populations on surfaces.

The proliferation of new transconjugants was highly predictable at intermediate stages of community succession, but stochastic at early and late stages. Small population sizes are susceptible to stochastic drift[52], which during range expansion can drive deleterious genetic variants to fixation[53]. We observed small and unpredictable numbers of transconjugant lineages at the expansion frontier at both early and late antibiotic administration times. At early administration times, a few "lucky" sensitive lineages were able to obtain the AR-encoding plasmid and benefit from antibiotic administration to colonize the expansion frontier. Likewise, at later stages of community succession, a few AR lineages had drifted to the expansion frontier and proliferated upon antibiotic administration (Supplementary Fig. 4). This is an example of how the persistence of a deleterious mutation drifting at the frontier of a range expansion can proliferate when environmental conditions change[54]. The lineage diversity of an expanding population sets the

basis for its subsequent adaptation to novel conditions[55], a factor that can determine the probability of AR spread into different environments. Our results suggest that the timing of antibiotic administrations can be important for controlling the heterogeneity (i.e. lineage diversity) of AR, which should be a key determinant for predicting the potential threat of a community carrying AR genes[56–59]. The number of unique transconjugant cell lineages followed the same unimodal trend as population sizes, where lineage diversity peaked at intermediate successional stages. However, the size and genetic diversity of the transconjugant population decoupled over time after antibiotic administration (Supplementary Fig. 3), where some lineages drifted to extinction while others kept growing. These observations are conceptually similar to those by Stevenson et al.[60], who showed that mercury resistance encoded by conjugative plasmids spreads predominantly horizontally in the absence of mercury stress (here time before antibiotic administration), while resistance spreads predominantly vertically via clonal expansion in the presence of mercury stress (plasmid spread after antibiotic administration). Consistent with dynamics in range expansions under the strong effects of genetic drift[32], our findings indicate that the diversity of newly formed AR populations is determined by the time lapsed before antibiotic administration, but that this diversity of the AR lineages surviving the treatment decreases rapidly over time.

The counterbalancing effects of plasmid transfer and loss determine the time when antibiotic administration most effectively promotes the proliferation of plasmid-encoded AR. The transfer and loss rates of plasmids can offset the influence of plasmid fitness costs in the maintenance of AR[46,48,60]. For example, Lopatkin et al.[10] showed experimentally that nine common plasmids across six incompatibility groups can persist in microbial consortia in the absence of positive selection provided transfer rates are sufficiently high. Similarly, Porse et al.[61] found that plasmid loss rates offset the fitness cost of 14 plasmids found in E. coli strains commonly involved in urinary tract infections, driving AR persistence in those strains. We show that the interplay between these factors determines the spread of AR-encoding plasmids differently depending on the successional stage of the community. Very high plasmid transfer or very low loss rates can both lead to the preservation of plasmids at the expansion frontier over time, leading to maximal AR spread at later times of antibiotic administration (Fig. 5d, e). This finding confirms previous work showing that it only takes the presence of a highly proficient donor (one with high plasmid transfer and low plasmid loss rates) to maintain plasmids in adjacent poor recipient populations (those with low plasmid transfer and high plasmid loss rates)[8]. On the other end, very low plasmid transfer or very high plasmid loss rates rapidly purge the plasmid from the expansion frontier and prevent AR spread (Fig. 5d, e), and it is only when plasmid transfer and loss rates balance each other that maximal AR spread occurs at intermediate stages of community succession.

Plasmid loss rates are highly variable even among strains of the same species[46], and in complex communities, plasmid persistence in the absence of positive selection is associated with the proportion of highly proficient versus poor strains at maintaining and transferring the plasmid[62]. Intuitively, the presence of poor plasmid recipients is higher in complex communities, which could hamper the maintenance of AR. However, Kottara et al.[62] found that plasmid transfer and loss rates can offset the influence of the selective environment and of specific plasmid features to determine plasmid spread in soil microbial communities. Similar experiments to those shown here that track plasmid dynamics in more complex communities containing taxa with different plasmid transfer and loss rates would help understand how varying levels of plasmid transfer and loss rates influence the spread of plasmid-encoded AR in natural systems. We are yet aware that estimating the plasmid transfer probability is not trivial as numerous abiotic and biotic factors need to be considered such as nutrient level, pH, and temperature[63–65].

Especially in spatially structured environments, individual cells likely encounter different environmental conditions and thus have different transfer and loss rates at different locations.

Our results can aid the mechanistic understanding of the spread of plasmid-encoded AR on surfaces colonized by microbial communities while acknowledging that plasmid dynamics are likely far more complex in natural systems. The growing body of studies that identify high plasmid transfer or low plasmid loss rates as the primary mechanism for AR maintenance in the absence of antibiotic pressure (e.g., Lopatkin et al.[10]) suggests cessation of antibiotic use will not be sufficient to eradicate plasmid-encoded AR over prolonged periods of time. Because the timing of antibiotic administration can be a factor that varies under different circumstances[66–68], we believe it is important to understand the relationships between antibiotic administration times and the spread of AR. We thus believe that a better temporal understanding of the interplay between plasmid transfer and loss in more complex microbial communities is essential to better understand the problem of AR persistence in efforts to tackle the global AR crisis.

## Methods

### Bacterial strains and plasmid

We provide the genotypes of all the *P. stutzeri* strains used in this study in Supplementary Table 2. Detailed descriptions of the construction of our strains can be found elsewhere[37,40,69]. Briefly, the donor and recipient strains are identical except that they carry different isopropylthio-β-galactoside (IPTG)-inducible fluorescent protein-encoding genes on their chromosome. The donor strain contains a red fluorescent protein-encoding gene (*echerry*) while the recipient strain contains a green fluorescent protein-encoding gene (*egfp*)[40]. This enables us to distinguish and quantify the abundances of different strains when grown together[37–39]. The donor strain additionally carries an R388-derivative plasmid pAR145 (pSU2007 *aph::cat*-$P_{A1/04/03}$-*cfp*$^*$-$T_0$) that encodes for chloramphenicol resistance and an IPTG-inducible *ecfp* gene (encoding for blue fluorescent protein)[41,42].

### Range expansion experiments

We used a modified version of the range expansion experimental protocols reported elsewhere[32,37]. We first grew the pAR145 donor strain overnight with liquid lysogeny broth (LB) medium amended with chloramphenicol (25 µg mL$^{-1}$) to maintain pAR145 and the potential recipient strain overnight in LB medium without chloramphenicol. After growth, we adjusted the optical density at 600 nm ($OD_{600}$) of each overnight culture to 2.0 with 0.89% (w/v) sodium chloride solution, mixed the pAR145 donor and potential recipient cultures together at a volumetric ratio of 1:1, and deposited 1 µL aliquots of the mixture onto the surfaces of separate replicated LB agar plates amended with 0.1 mM IPTG. We then incubated the LB agar plates at room temperature. Note that in order to ensure appropriate growth rates for investigation, we used 30% LB agar plates (1.5% agar) to reduce the nutrient level and slow the expansion velocity. We administered chloramphenicol at 13 different time points after the onset of range expansion, which are: 0, 2, 4, 6, 12, 24, 36, 48, 60, 72, 84, 96, and 108 h. We administered chloramphenicol (the final concentration within the agar plate was 25 µg/mL) by depositing a total volume of 10 µL to each LB agar plate as four 2.5 µL point sources located ~2 mm away from the expansion frontier. This concentration of chloramphenicol completely represses the growth of plasmid-free cells under our experimental conditions. We performed five independent biological replicates for each chloramphenicol administration time.

### Image acquisition and analysis

We acquired confocal laser scanning microscopy (CLSM) images of our range expansions using a Leica TCS SP5 II confocal microscope (Leica Microsystems, Wetzlar, Germany). We used objectives ×5/0.12na (dry) and ×10/0.3na (dry) (Etzlar, Germany). We set the laser emission at 458 nm for the excitation of blue fluorescent protein, at 488 nm for the excitation of green fluorescent protein, and at 514 nm for the excitation of red fluorescent protein. We used an image frame size of 1024 × 1024 and a pixel size of 3.027 µm.

We analyzed the images in ImageJ (https://imagej.nih.gov/ij/) using Fiji plugins (v. 2.1.0/1.53c) (https://fiji.sc). We first auto-thresholded channel four to obtain outlines of the expansion areas using the 'Otsu dark' function and used this information to quantify expansion size and signals at the expansion frontier. We next auto-thresholded channel three (the green channel) using the 'Intermodes dark' function, followed by applying the functions "Fill Holes" and "Despeckle" to remove noise. To segment and extract the red channel, we auto-thresholded channel five using the "Huang dark" function followed by the same noise-removing steps. We used the same approach to quantify the plasmid load by segmenting channel one (the blue channel). We multiplied the binarized blue channel with the red channel to obtain the signal for the pAR145 donor (composite magenta color) or the blue channel with the green channel to obtain the signal for transconjugants (composite cyan color). The composite channels for magenta and cyan register all the signals of either pAR145 donors or transconjugants across the whole expansion area. Next, we multiplied the composite channels with the extracted expansion outline to obtain the donor or transconjugant at the expansion frontier. Because all the images are binary images, the outline image is a "ring" that only contains values of 255 at the periphery, while the rest are all 0. Therefore, by multiplying the outline image with the composite channel of magenta or cyan, we are able to capture signals lying at the expansion frontier. We finally applied the 'analyze particle' function to obtain counts and sizes of desired objects such as the expansion size, frequency of transconjugants at the expansion frontier, and the ratio of two strains at the expansion frontier.

### Quantification of the global intermixing index

We quantified spatial intermixing (referred to as the intermixing index) between strains from the CLSM images using Fiji plugins (v. 2.1.0/1.53c) (https://fiji.sc). We first cropped all the images to squares and applied the 'Intermodes dark' function to channel three (green channel) followed by the "Despeckle" function twice to remove noise. We then used the Sholl analysis plugin[70] on the binarized channel three to calculate the number of intersections between the background and information-containing parts of the image. We next extracted data over defined ranges (for Fig. 2b, between radii of 2500 and 5000 µm; for Fig. 3c, between radii of 2000 and 4000 µm). We excluded radii <2500 or 2000 µm for two reasons: first, they do not accurately capture the spatial features caused by the range expansion process (i.e., they capture the inoculation area). Second, fluorescent signals at smaller radii are difficult to precisely resolve, thus creating noise. To obtain the global intermixing index, we summed the individual intermixing indices at 10 µm radial increments within the desired ranges and then normalized the sum by the number of radial increments that contained non-zero values. We quantified individual intermixing following the descriptions provided elsewhere[37].

### Relative growth rate measurements

We used a colony collision assay as described elsewhere[71] to measure the relative growth rate, or plasmid cost, of plasmid-carrying and plasmid-free strains in spatially structured environments. We first grew monocultures of the pAR145-carrying and pAR145-free strains independently and adjusted the $OD_{600}$ of each overnight culture to 2.0 with 0.89% (w/v) sodium chloride solution. We then used a pipetting robot to place two 1 µL drops, one of which was the pAR145 donor and the other the potential recipient, 3 mm apart from each other onto replicated LB agar plates (1.5% agar). We next incubated the LB agar plates at room temperature for 96 h to allow the drops to form colonies and the colonies to collide with each other. We then estimated the

relative growth rates of strains based on the arc of the collision boundary between the two corresponding colonies and the radii of the colonies using Eq. (1) below:

$$R_b = l \frac{v_1 v_2}{|v_1^2 - v_2^2|} = l \frac{1+s}{s(2+s)} \tag{1}$$

where $l$ is the distance between two colonies, $s$ is the selective advantage or the cost that pAR145 confers, $v_1$ is the expansion velocity of the pAR145-free colony that is growing faster, and $v_2$ is the expansion velocity of the pAR145-carrying colony that is growing slower. $R_b$ is the radius of the circle generated by the arc at the boundary, and knowing $R_b$ and $l$ is sufficient to derive $s$. We quantified $R_b$ and $l$ for 4 replicates using Adobe Illustrator (version 27.0.1) to manually draw lines and circles and to extract values. Image scale can differ among replicates, but since $R_b$ and $l$ are proportional in one image, $s$ will not be affected. We provide all of the data in Supplementary Table 1

### Individual-based computational modeling

We customized a spatially explicit individual-based computational model to mimic our experimental system using the CellModeller 4.3 framework[43]. CellModeller is a Python-based, open-source platform for modeling large-scale multi-cellular systems, such as biofilms, plant tissue, and animal tissue. We modeled individual rod-shaped bacterial cells as three-dimensional capsules that grow by extending their length. Capsules experience viscous drag and cannot grow into one another. As they grow, cells add a constant volume until they reach a critical size where they then divide into two daughter cells, ensuring cell size homeostasis. In CellModeller, each cell is abstracted as a computational object referred to as a cellState (cs) that contains all the information regarding that individual cell, including its spatial position (pos[$x, y, z$]), rotational orientation (dir[$x, y, z$]), cell length (length), growth rate (growthRate), and cell type (cellType). The cell-type is an arbitrary label that allows us to simulate different cellular behaviors. Our model contains four cell types: cellType 0 simulates a potential recipient cell colored green; cellType 1 simulates a plasmid donor cell colored magenta; cellType 2 simulates the plasmid-free status of cellType1, colored red; and cellType 3 simulates the plasmid-carrying status of cellType 0, colored cyan. These four cellTypes allow us to distinguish each type from the others and record information, for example, on spatial positioning during the simulation.

In CellModeller, individual cells are modeled as cylinders of length $l$ capped with hemispheres that result in a capsule shape, with both hemispheres and the cylinder having a radius $r$. At each simulation step, a cell increases in length based on its growth rate parameter, which is physically constrained by the other cells in its physical proximity. In this work, we initiated cells to have $r = 0.04$ and $l = 2$ and set cells to divide when their length reaches the critical division length $l_{div}$ with Eq. (2) below:

$$l_{div} = l_0 + G(\triangle, \sigma) \tag{2}$$

where $l_0$ is the initial cell length at birth and $G$ is a random Gaussian distribution with mean $\triangle = 2$ and standard deviation $\sigma = 0.45$. Therefore, when a cell divides, the two daughter cells are initiated with $l_{div}/2$ and a new target division length is assigned to each daughter cell calculated from the equation above. The addition of constant mass has been found to accurately model bacterial division while maintaining cell size homeostasis as described elsewhere[72].

We modified our model to integrate plasmid transfer and loss. As part of the biophysics in CellModeller, physical contacts between cells are calculated at each step to minimize any overlap between cells[43]. We altered the code such that each cell kept track of its contacts, thus allowing us to model plasmid transfer when cells were in contact. This function is activated by setting the argument 'compNeighbours = True' when initiating the biophysical model. When plasmid donor and recipient cells were in contact, we applied a constant probability per unit time of plasmid transfer. For all figures except for Fig. 5e, we applied a constant probability ($P_c = 0.002$) for plasmid transfer and varied the probability of plasmid loss to investigate the interplay between the two. We applied "antibiotics" at six different time steps: 0, 100, 200, 400, 600, and 800. We modified the self-defined function of updating cell status where we only allowed plasmid-carrying cells to continue growing after "antibiotic treatment". Plasmid transfer and loss can occur during the entire simulation regardless of whether "antibiotics" were applied or not.

We ran all high-resolution simulations in parallel on Piz Daint, a supercomputer located at the Swiss National Supercomputing Center (CSCS). We loaded the modules "daint-mc" and the high throughput scheduler "GREASY" for high throughput simulations. We used the Slurm workload manager to submit jobs via the command "sbatch". We generated job scripts following the template scripts on Slurm jobscript generator.

### Analysis of simulation data

We extracted and generated simulation data on Piz Daint and converted all necessary information from pickle files to csv files. We then performed all statistical analyses using R Studio Version 1.3.1073 (https://www.rstudio.com). We used the two-sample two-sided Welch test for all pair-wise comparisons, and we therefore did not make any assumptions regarding homogeneity of variances among our datasets. To identify the trend of unimodality, monotonic increasing, monotonic decreasing, or flat, we obtained the maximum average value and compare it with the earliest treatment time point and the latest treatment time point. If the maximum value emerges at an intermediate time point, then the values at the beginning and end should be significantly lower than the maximum observed value, thus indicating a unimodal trend. If the maximum value is only significantly higher than the earliest treatment point, then it indicates a monotonically increasing trend. If it is only significantly higher than the latest treatment point, then it indicates a monotonically decreasing trend. Finally, if it is not significantly higher than either of these two endpoints, then it indicates a flat trend. All sample sizes ($n$) reported in the results are the number of independent biological replicates.

### Reporting summary

Further information on research design is available in the Nature Portfolio Reporting Summary linked to this article.

## Data availability

All data generated in this study have been deposited in the Eawag Research Data Institutional Collection (ERIC) repository (https://opendata.eawag.ch/) at the following https://doi.org/10.25678/0008EB.

## Code availability

All codes used in this study are publicly available in the Eawag Research Data Institutional Collection (ERIC) repository (https://opendata.eawag.ch/) at the following https://doi.org/10.25678/0008EB.

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

## Acknowledgements

We thank Stuart Dennis for assistance with the simulation performance on Piz Daint. We acknowledge access to Piz Daint at the Swiss National Supercomputing Centre (CSCS) under Eawag's share with the project ID em09. We thank Daniel Angst for assistance with the pipetting robot at ETH Zürich. Y.M. was supported by grants from the Swiss National Science Foundation (31003A_176101 and 310030_207471) awarded to D.R.J. J.R. was supported by an Early PostDoc Mobility grant from the Swiss National Science Foundation (P2EZP3_199849) awarded to J.R.

## Author contributions

Y.M. and D.R.J. conceived and developed the main research question. Y.M. and D.R.J. designed the laboratory experiments. All authors designed the in silico experiments. Y.M. performed the experiments and individual-based model simulations. All authors analyzed and interpreted the data. All authors wrote and revised the manuscript. All authors reviewed and approved the final version of the manuscript.

## Competing interests

The authors declare no competing interests.
