## [Peer Review File · Nature Communications]

Timing of antibiotic administration determines the spread of plasmid-encoded antibiotic resistance during microbial range expansionReviewer #1 (Remarks to the Author):

The manuscript "Timing of antibiotic administration drives population dynamics of AR-encoded plasmids during range expansions" examines how the timing of drug administration affects the spread of antibiotic resistance genes encoded in plasmids within microbial communities. The authors use a combination of experimental approaches and computational modeling to explore this question, and their findings provide insights into the dynamics of plasmid-encoded resistance gene spread in surface-associated microbial communities. Interestingly, their results suggest that the most effective time for promoting the spread of plasmid-encoded antibiotic resistance is at intermediate stages of community succession. This finding challenges the conventional wisdom of using antibiotics with a hit-early, hit-hard strategy (as stated by Paul Erlich more than a century ago), therefore highlighting the importance of considering the temporal dynamics of antibiotic administration when tackling the global problem of antibiotic resistance.

The experimental model system comprised a Petri dish containing a solid medium with antibiotics added at multiple time points. The microcosm was inoculated with a mixed bacterial community consisting of strains of *Pseudomonas stutzeri* and a conjugative plasmid that encodes both chloramphenicol resistance and an IPTG-inducible fluorescent gene. The spatial expansion of the microbial community was tracked over time using fluorescent markers, and the spread of antibiotic-resistant plasmids was assessed using image analysis. Using this approach, the authors found that the frequency of transconjugant cells, which are cells that have acquired the plasmid through horizontal gene transfer, increased when chloramphenicol was administered at intermediate times after the onset of range expansion. Furthermore, the ability of the consortia to expand during chloramphenicol treatment was also found to be highest at intermediate times.

The authors also used an individual-based computational model to investigate the factors driving the spread of resistance genes in microbial communities. The model is implemented using CellModeler and allowed the authors to simulate the dynamics of plasmid transfer and loss, as well as the growth and movement of individual bacterial cells. The results obtained from the computational model are in agreement with the experimental results, suggesting that the increased frequency of transconjugant cells observed in the experiments at intermediate antibiotic administration times was largely due to the local proliferation of individual transconjugants, rather than more transfer events. Additionally, the authors found that the timing of antibiotic administration after the onset of range expansion had a unimodal relationship with the number of transconjugant lineages, the frequency of transconjugant cells at the expansion frontier, and the mean size of transconjugant lineages. The authors also found that the increased frequency of transconjugant cells was largely associated with the local proliferation of individual transconjugants, rather than more transfer events.

Overall, this study presents a well-designed and well-executed investigation that provides valuable insights into the temporal dynamics of plasmid-encoded antibiotic resistance spread in microbial communities. The findings underscore the importance of understanding the relationships between antibiotic administration times and the spread of antimicrobial resistance, and therefore this study has the potential to be of interest to a wide range of researchers and clinicians concerned with antimicrobial resistance and could help shape future research in this area.

Reviewer #2 (Remarks to the Author):

The work by Ma et al. "Timing of antibiotic administration determines the spread of plasmid-encoded antibiotic resistance during microbial range expansion" studies the effect of timing of antibiotic exposure on plasmid dispersal and loss in *P. stutzeri*. The study utilizes a range expansion setup where the identification of donor, recipient and transconjugant strains is based on fluorescence markers which enables measuring their frequencies during antibiotic exposure. Further, the plasmid transfer dynamics after varying duration before antibiotic treatment is modelled based on experimental data. The main finding of this manuscript is that the timing of the antibiotic treatment determines the frequency of the plasmid-carrying population at the expansion

border; and more specifically, early exposure does not give enough time for the plasmid to transfer to plasmid-free cells and formation of transconjugants, whereas longer time allows both new transconjugants to form and plasmid loss from original hosts and transconjugants. Interestingly, intermediate timing of antibiotic exposure seems to result in highest frequencies of plasmid carrying populations.

The manuscript well describes these plasmid dynamics during range expansion and the evolutionary effects driving these dynamics are sufficiently covered in the text. Overall, the manuscript is very well-written and clearly describes the defined research questions and the methodology used for exploring the questions.

But what was not exactly clear is that when growth rates in plasmid-carrying and plasmid-free cells were measured, was this done only in the absence of antibiotics? I would assume that the growth rate of the plasmid-carrying cells with or without antibiotics affects the growth rate.

Further, was this taken into account in the modelling? If yes, please indicate clearly in the text.

Otherwise this paper has an exciting approach to study and predict how timing of antibiotic exposure affect the plasmid dispersal which is an important aspect to consider when applying antibiotic therapies, however, keeping in mind that in natural communities the plasmid dynamics are presumably much more complex.

We thank yourselves and both reviewers for your time and effort and for your positive assessment of our work. We have now modified our manuscript accordingly regarding the one concern raised by Reviewer 2. We believe that addressing this concern was important to prevent confusion to the reader, and we therefore thank Reviewer 2 for pointing it out.

All line numbers refer to those in the revised manuscript. Reviewer comments are indicated in normal font. Author responses are indicated in blue font. Specific changes to the text are indicated in *blue italic font*.

Sincerely and on behalf of the co-authors,

Dave Johnson

Reviewer #1 (Remarks to the Author):

The manuscript "Timing of antibiotic administration drives population dynamics of AR-encoded plasmids during range expansions" examines how the timing of drug administration affects the spread of antibiotic resistance genes encoded in plasmids within microbial communities. The authors use a combination of experimental approaches and computational modeling to explore this question, and their findings provide insights into the dynamics of plasmid-encoded resistance gene spread in surface-associated microbial communities. Interestingly, their results suggest that the most effective time for promoting the spread of plasmid-encoded antibiotic resistance is at intermediate stages of community succession. This finding challenges the conventional wisdom of using antibiotics with a hit-early, hit-hard strategy (as stated by Paul Erlich more than a century ago), therefore highlighting the importance of considering the temporal dynamics of antibiotic administration when tackling the global problem of antibiotic resistance.

The experimental model system comprised a Petri dish containing a solid medium with antibiotics added at multiple time points. The microcosm was inoculated with a mixed bacterial community consisting of strains of *Pseudomonas stutzeri* and a conjugative plasmid that encodes both chloramphenicol resistance and an IPTG-inducible fluorescent gene. The spatial expansion of the microbial community was tracked over time using fluorescent markers, and the spread of antibiotic-resistant plasmids was assessed using image analysis. Using this approach, the authors found that the frequency of transconjugant cells, which are cells that have acquired the plasmid through horizontal gene transfer, increased when chloramphenicol was administered at intermediate times after the onset of range expansion. Furthermore, the ability of the consortia to expand during chloramphenicol treatment was also found to be highest at intermediate times.

The authors also used an individual-based computational model to investigate the factors driving the spread of resistance genes in microbial communities. The model is implemented using CellModeler and allowed the authors to simulate the dynamics of plasmid transfer and loss, as well as the growth and movement of individual bacterial cells. The results obtained from the computational model are in agreement with the experimental results, suggesting that the increased frequency of transconjugant cells observed in the experiments at intermediate antibiotic administration times was largely due to the local proliferation of individual transconjugants, rather than more transfer events. Additionally, the authors found that the timing of antibiotic administration after the onset of range expansion had a unimodal relationship with the number of transconjugant lineages, the frequency of transconjugant cells at the expansion frontier, and the mean size of transconjugant lineages. The authors

also found that the increased frequency of transconjugant cells was largely associated with the local proliferation of individual transconjugants, rather than more transfer events.

Overall, this study presents a well-designed and well-executed investigation that provides valuable insights into the temporal dynamics of plasmid-encoded antibiotic resistance spread in microbial communities. The findings underscore the importance of understanding the relationships between antibiotic administration times and the spread of antimicrobial resistance, and therefore this study has the potential to be of interest to a wide range of researchers and clinicians concerned with antimicrobial resistance and could help shape future research in this area.

We thank the reviewer for their positive assessment of our work. The reviewer's summary and interpretation of our work is entirely accurate. The reviewer did not raise any concerns or provide any suggestions for improvement.

Reviewer #2 (Remarks to the Author):

The work by Ma et al. "Timing of antibiotic administration determines the spread of plasmid-encoded antibiotic resistance during microbial range expansion" studies the effect of timing of antibiotic exposure on plasmid dispersal and loss in *P. stutzeri*. The study utilizes a range expansion setup where the identification of donor, recipient and transconjugant strains is based on fluorescence markers which enables measuring their frequencies during antibiotic exposure. Further, the plasmid transfer dynamics after varying duration before antibiotic treatment is modelled based on experimental data. The main finding of this manuscript is that the timing of the antibiotic treatment determines the frequency of the plasmid-carrying population at the expansion border; and more specifically, early exposure does not give enough time for the plasmid to transfer to plasmid-free cells and formation of transconjugants, whereas longer time allows both new transconjugants to form and plasmid loss from original hosts and transconjugants. Interestingly, intermediate timing of antibiotic exposure seems to result in highest frequencies of plasmid carrying populations.

The manuscript well describes these plasmid dynamics during range expansion and the evolutionary effects driving these dynamics are sufficiently covered in the text. Overall, the manuscript is very well-written and clearly describes the defined research questions and the methodology used for exploring the questions.

We thank the reviewer for their positive assessment of our work. The reviewer's summary and interpretation of our work is entirely accurate.

But what was not exactly clear is that when growth rates in plasmid-carrying and plasmid-free cells were measured, was this done only in the absence of antibiotics? I would assume that the growth rate of the plasmid-carrying cells with or without antibiotics affects the growth rate. Further, was this taken into account in the modelling? If yes, please indicate clearly in the text.

This is a great point that we should have been clearer about in our original submission. While we quantified and reported the relative growth rate of the plasmid-carrying to -free cells in the absence of antibiotics, we did not report the relative growth rate in the presence of antibiotics. The answer is quite simple; the antibiotic concentration that we used was well above the MIC and completely prevented observable growth of the plasmid-free cells. Thus, the relative growth rate of the plasmid-carrying to -free cells in the presence of antibiotics is approximately infinity. We now state that the concentration of antibiotics that we used

completely represses growth of the plasmid-free cells in the following lines of the revised manuscript.

Lines 164-165: *“Note that the chloramphenicol concentration that we applied prevents further growth of plasmid-free cells.”*

Lines 473-475: *“This concentration of chloramphenicol completely represses the growth of plasmid-free cells under our experimental conditions.”*

We also state this for the individual-based computational model in the following lines of the revised manuscript. These lines were already present in our original submission, and they are therefore not marked in the revised version.

Lines 598-600: *“We modified the self-defined function of updating cell status where we only allowed plasmid-carrying cells to continue growing after “antibiotic treatment.”*

Otherwise this paper has an exciting approach to study and predict how timing of antibiotic exposure affect the plasmid dispersal which is an important aspect to consider when applying antibiotic therapies, however, keeping in mind that in natural communities the plasmid dynamics are presumably much more complex.

We again thank the reviewer for their positive assessment of our work. We agree that plasmid dynamics are likely far more complex in nature and now state this point explicitly in the following lines of the revised manuscript.

Lines 428-430: *“Our results can aid the mechanistic understanding of the spread of plasmid-encoded AR on surfaces colonized by microbial communities, while acknowledging that plasmid dynamics are likely far more complex in natural systems.”*